# MEEMD Decomposition–Prediction–Reconstruction Model of Precipitation Time Series

**DOI:** 10.3390/s22176415

**Published:** 2022-08-25

**Authors:** Yongtao Wang, Jian Liu, Rong Li, Xinyu Suo, Enhui Lu

**Affiliations:** 1State Key Laboratory of Advanced Design and Manufacture for Vehicle Body, Hunan University, Lushan South Road, Yuelu District, Changsha 410082, China; 2Guizhou Institute of Water Resources Science, Guiyang 550002, China; 3School of Mechanical Engineering, Yangzhou University, Yangzhou 225012, China

**Keywords:** improved overall mean empirical modality (MEEMD), particle swarm optimization support vector machine (PSO-SVM), convolutional neural network (CNN), recurrent neural network (RNN), improved overall mean empirical modality decomposition–prediction–reconstruction model (MDPRM)

## Abstract

To address the problem of low prediction accuracy of precipitation time series data, an improved overall mean empirical modal decomposition–prediction–reconstruction model (MDPRM) is constructed in this paper. First, the non-stationary precipitation time series are decomposed into multiple decomposition terms by the improved overall mean empirical modal decomposition (MEEMD). Then, a particle swarm optimization support vector machine (PSO-SVM) and convolutional neural network (CNN) and recurrent neural network (RNN) models are used to make predictions according to the characteristics of different decomposition terms. Finally, the prediction results of each decomposition term are superimposed and reconstructed to form the final prediction results. In addition, the application is carried out with the summer precipitation in the Wujiang River basin of Guizhou Province from 1961 to 2018, using the first 38 years of data to train MDPRM and the last 20 years of data to test MDPRM, and comparing with a feedback neural network (BP), a support vector machine (SVM), a particle swarm optimization support vector machine (PSO-SVM), a convolutional neural network (CNN), and a recurrent neural network (RNN), etc. The results show that the mean relative error (*MAPE*) of the proposed MDPRM is reduced from 0.31 to 0.09, the root mean square error (*RMSE*) is reduced from 0.56 to 0.30, and the consistency index (α) is significantly improved from 0.33 to 0.86, which has a higher prediction accuracy. Finally, the trained MDPRM predicts the average summer precipitation in the Wujiang River basin from 2019 to 2028 to be 466.42 mm, the minimum precipitation in 2020 to be 440.94 mm, and the maximum precipitation in 2024 to be 497.94 mm. Based on the prediction results, the agricultural drought level is evaluated using the Z index, which indicates that the summer is normal in the 10-year period. The study provides technical support for the effective guidance of regional water resources’ allocation and scheduling and drought mitigation.

## 1. Introduction

Precipitation prediction can scientifically and rationally guide the planning, development, and distribution of regional water resources, reduce flood and drought disaster losses, and alleviate water shortages, etc. The prediction results can provide reference for water supply and demand analysis, water supply planning, and water resources management. By analyzing the precipitation data of the Wujiang River basin, it is concluded that the average precipitation in the summer (June–August) accounts for 45.16% of the annual average precipitation in 58 years from 1961 to 2018, which is the main part of the annual precipitation and the months in which extreme precipitation is concentrated. Therefore, the correct understanding and grasp of the development of extreme precipitation events in Guizhou Province seasonal time scale change characteristics and accurate prediction, for effective guidance of the summer flood period of water resources scheduling management and safe flood control work is of great significance. Precipitation is a weakly correlated highly complex nonlinear dynamical system with interannual variation not in a fixed cycle, but with various time-scale changes and local fluctuations, making it difficult and less accurate to predict precipitation in the medium and long term. Most scholars have predicted regional precipitation by single methods such as the grey system theory, fuzzy mathematics, artificial neural network, wavelet transform, and support vector machine, which greatly affect the medium- and long-term prediction of non-stationary time series due to the different characteristics and applicability conditions of each model method and the large uncertainty of parameter estimation. However, the improved integrated empirical modal decomposition (MEEMD) of precipitation data can more accurately identify the hierarchical features in the precipitation data, effectively reduce the interference between the decomposition terms, reduce the complexity of prediction, and select a suitable prediction method for each decomposition term to improve the overall prediction accuracy and prediction efficiency of the combined prediction methods with more obvious advantages.

Related studies conducted at home and abroad are as follows: Zhang et al., used envelope entropy (PE) to improve the pseudo-components of integrated empirical modal decomposition (EEMD) to reduce the non-smoothness of time series in order to improve the prediction accuracy of annual runoff in the lower Yellow River. A new coupled model of improved integrated empirical modal decomposition autoregressive integrated moving average (MEEMD-ARIMA) is constructed and applied to the runoff prediction of the lower Yellow River. The results show that the model has high accuracy and outperforms the complementary EEMD-autoregressive integrated moving average (CEEMD-ARIMA) model or the complementary EEMD-autoregressive integrated moving average (EEMD-ARIMA) model, providing a new idea and method for annual runoff prediction [1]; Wu et al., proposed a new coupled model based on an improved grey BP neural network (BPNN) and MEEMD-ARIMA for predicting wave energy as an integrated model. A case study was conducted using wind and wave data from actual ocean measurements to verify the validity of the proposed wave energy prediction model [2]. Wang et al., used the China Shanghai Futures Exchange (CSFE) to predict wave energy by introducing the mirror image expansion (ME), the empirical mode decomposition (EMD), the cuckoo search (CS) algorithm, and the Elman neural network. (SFE) gold futures AU0 price data from 29 August 2013 to 18 October 2018, a mirror image extension improved integrated empirical mode decomposition (MEEMD)-CS-Elman model is constructed. The empirical results of the model show that Elman combined with EMD outperforms single Elman, and by applying the proposed model investors can make more scientific and accurate decisions and better reduce investment risks [3]. Song et al., proposed a MEEMD-sparse autoencoder (SAE)-Elman ultra-short-term wind portfolio forecasting method, which improves the EMD step in the decomposition process of the MEEMD algorithm by using cubic triangular Hermitian interpolation. The MEEMD is used to decompose the raw wind power data into IMF components, obtain the sample entropy values of each IMF component, and combine them to reduce the computational effort in modeling. Then, a deep-learning SAE prediction model is built for the high-frequency components and an Elman neural network prediction model is introduced for the low-frequency components. Finally, the wind power prediction values are obtained by superimposing the previous data and analyzing the errors. By comparing several prediction models in the example, the prediction accuracy of the combined prediction model is improved and can be applied to ultra-short-term prediction of wind power [4]. Xie et al., proposed a new method based on the MEEMD, approximate entropy, and a weighted least square support vector machine (WLS-SVM). The method focuses on the chaotic sequences from time-frequency analysis and improves the model performance as follows: firstly, the time series is decomposed into a series of subsequences with significantly different complexity using MEEMD. Then, a new subsequence is generated by combining subsequences of similar complexity with the approximate entropy method, which can effectively concentrate the component feature information and reduce the computational scale. Finally, a WLS-SVM prediction model is built for each new subsequence. Meanwhile, the input dimension and optimal parameters of the model are selected by using the phase space reconstruction theory and the grid search method and then each prediction value is superimposed as the final prediction result. Experiments were conducted with landslide deformation data from Damba and compared with a wavelet neural network, a support vector machine, a least square support vector machine, and various combination schemes. The experimental results show that the algorithm has high prediction accuracy. Even in the period of rapid fluctuation of landslide deformation, it can ensure better prediction and it can control the residual values better and effectively reduce the error interval [5]. Lin et al., used the MEEMD method combined with a long short-term memory neural network (LSTM) for precious metal price prediction in order to achieve a higher prediction accuracy of precious metal prices. Multiscale reciprocal entropy (MPE) analysis shows that MEEMD has better decomposition than EEMD. Then, each eigenmodal component (IMF) obtained from MEEMD is input into LSTM for prediction. Finally, the prediction values of each IMF are added to obtain the final prediction results. The improved integrated empirical modal decomposition-long and short-term memory neural network (MEEMD-LSTM) improves the prediction performance compared with the traditional multilayer perceptron neural network (MLP), the support vector regression (SVR), and the combined prediction model super learner (SL). The best prediction performance of MEEMD-LSTM was comprehensively and statistically demonstrated using a multi-span model confidence set (MCS) test. Furthermore, the study shows that the model still shows better prediction accuracy both at different ratios of the training and test sets and at different periods of the business cycle. This competitive precious metal price forecasting model is a promising technique for government agencies, investors, and related companies [6]. Yang et al., proposed a novel hybrid model that combines MEEMD and LSTM and is optimized by an improved whale optimization algorithm (IWOA). The model is based on the nonlinear and non-smooth properties of carbon prices. First, the original carbon price is decomposed into nine IMF components and one residual using the MEEMD model. Then, the random forest method is applied in the LSTM neural network to determine the input variables for each IMF and residual. Finally, the IWOA-optimized LSTM model is used to predict the carbon price. The hybrid model proposed in this study achieved higher prediction performance than the other 11 benchmark models in predicting carbon prices in Beijing, Fujian, and Shanghai. The model proposed in the study provides a novel and effective carbon price forecasting tool for the government and enterprises [7]. Yang et al., proposed a multi-factor carbon price forecasting method MEEMD-LSTM. The method is based on MEEMD decomposing data as potential input variables into LSTM neural networks for forecasting and a production rule-based machine inference system can automatically search and optimize parameters of the LSTM to further improve the prediction results. The experimental results show that the proposed method has better prediction results, robustness, and adaptability compared with the LSTM model without MEEMD decomposition and the single-factor MEEMD-LSTM method [8]. The characteristics of the above combined prediction methods are shown in Table 1, and in general, the MEEMD decomposition–prediction–reconstruction method is an advanced method for predicting non-smooth and nonlinear signals and has achieved better prediction results in annual runoff, wave energy, gold futures AU0 price data, wind power, landslide deformation, precious metal prices, and carbon prices, but a lot of research is still needed in precipitation time series data prediction.

To this end, MDPRM is proposed in this paper and applied to the summer precipitation prediction in the Wujiang River basin. The improved integrated empirical modal decomposition (EEMD) algorithm is constructed by improving the integrated empirical modal decomposition (MEEMD) through the randomness detection based on the permutation entropy (PE). First, the original summer precipitation signal x in the Wujiang River basin for 58 years (1961–2018) is decomposed into five layers of IMF1-IMF4 with different frequencies and the residual term RS5, and the errors of each decomposition term are checked. Second, the prediction methods such as PSO-SVM, CNN, and RNN are preferred to carry out simulations for different decomposition terms that meet the requirements. Third, the predicted values of each decomposition term are combined to form. Finally, the constructed MDPRM is used to predict the summer precipitation in the Wujiang River basin for the next 10 years (2019–2028) and to evaluate the drought situation. The results show that the MDPRM constructed in this paper significantly improves the prediction accuracy and outperforms BP, SVM, PSO-SVM, CNN, and RNN, etc. The MDPRM fully combines the ability of MEEMD to accurately identify multi-level features of nonlinear and non-smooth signals and preferably selects PSO-SVM, CNN, RNN, and other prediction models for the combination of prediction, which provides a new method to improve the regional precipitation. It provides a new way of thinking and a method to improve the accuracy of regional precipitation prediction.

## 2. MEEMD Principle

The MEEMD algorithm is constructed by improving the EEMD based on the randomness detection of PE. For non-stationary signals, the MEEMD method decomposition steps are as follows [9,10,11]:
(1)To the original signal S(t), add white noise signals $n_{i}(t)$ and $-n_{i}(t)$ with zero mean, respectively, i.e.,:(1)$$S_{i}^{+}(t)=S(t)+a_{i}n_{i}(t)$$
(2)$$S_{i}^{-}(t)=S(t)-a_{i}n_{i}(t)$$
where $n_{i}(t)$ denotes the added white noise signal, $a_{i}$ denotes the amplitude of the added noise signal, i=1,2,⋯,Ne and, Ne denotes the number of added white noise pairs. The EMD decomposition is performed on $S_{i}^{+}(t)$ and $S_{i}^{-}(t)$, respectively, to obtain the first-order IMF component series, $\left\{I_{i1}^{+}(t)\right\}$ and $\left\{I_{i1}^{-}(t)\right\}(i=1,2,\cdots ,Ne)$. The components $I_{1}(t)=\frac{1}{2N}\sum_{i=1}^{Ne}\left[I_{i1}^{+}(t)+I_{i1}^{-}(t)\right]$ obtained above are integrated and averaged. $I_{1}(t)$ is checked for anomalies. The PE of the time series is calculated according to Equation (3), and, if the PE value of the signal is greater than $\theta_{0}$, it is considered as an anomalous signal, otherwise it is considered as approximately smooth. After several experiments, it is found that $\theta_{0}$ is more appropriate to take 0.55~0.6, and 0.6 is taken in this paper.The PE of the time series $\left\{x(i),i=1,2,\cdots ,N\right\}$ can be defined in the form of Shannon entropy as:(3)$$Hp(m)=-\sum_{g=1}^{m}P_{g}\ln P_{g}$$
where m is the embedding dimension and $P_{g}$ is the frequency of occurrence of each symbol sequence.(2)If $I_{1}(t)$ is abnormal, continue (1) until the IMF component $I_{p}(t)$ is not an abnormal signal.(3)Separate the first *p* − 1 components that have been decomposed from the original signal, i.e.,:(4)$$r(t)=S(t)-\sum_{j=1}^{p-1}Ij(t)$$(4)Then, the EMD decomposition is performed on the remaining signal r(t). The decomposition steps of EMD are as follows [12,13]:
➀For any signal s(t), first determine the extreme value points on s(t) and then connect all the extreme value points and the extreme small value points with a curve, respectively.➁The two curves are used as the upper and lower envelopes of s(t), respectively. The mean value of each point of the upper and lower envelopes is m, and the difference between s(t) and m is h, then h=s(t)−m.➂Consider h as the new s(t), repeat the above operation, when m meets the condition of zero or near zero, h is small enough or one of the h monotonic functions, record $c_{1}=h$. Consider $c_{1}$ as IMF1(t) and then record as $s(t)-c_{1}=r_{1}(t)$.➃Consider $r_{1}(t)$ as a new s(t) and repeat the above process. Obtain IMF2(*t*), $c_{2}$,$r_{2}$, IMF3(*t*), $c_{3}$, $r_{3}$,…, in turn. The process terminates when $c_{n}$ or $r_{n}(t)$ satisfies the given termination condition (the residual term is small enough or becomes a monotonic function) and the decomposition is obtained as:(5)$$s(t)=\sum_{i=1}^{n}{IMF}_{i}(t)+r_{n}(t)$$
where $r_{n}(t)$ is called the residual term.(5)All the obtained IMF components are arranged from high frequency to low frequency to obtain the decomposition terms IMF1, IMF2, …, IMFN, and the residual term RSN.

## 3. MDPRM Design and Forecasting Methodology

### 3.1. MDPRM Design

Due to the complexity of precipitation prediction, especially because the high-frequency components are random variables and non-stationary signals, it is difficult to make accurate predictions using conventional prediction methods. In this paper, we propose a new combined MDPRM precipitation prediction model, as shown in Figure 1. The specific steps of MDPRM modeling are as follows [14,15,16]:Step 1:data pre-processing: complete data storage management, basic analysis of mean and variance interrelationship of data, and calculation of correlation coefficients between data; eliminating obviously wrong data through analysis and deriving usable sample set on this basis; decomposing the usable sample set using MEEMD.Step 2:training and testing: the prediction models are built, trained, and tested using PSO-SVM, CNN, and RNN, respectively, according to the signal characteristics of each decomposition term.Step 3:prediction of each decomposition term: on the basis of completing training and testing, the constructed prediction methods are used to predict the components of different frequencies respectively.Step 4:combined reconstruction of prediction values: the final prediction results are obtained by combining the prediction values of each component, and the evaluation and feedback of the prediction results are completed.


### 3.2. Forecasting Methods

Since precipitation is subject to the combined effects of many physical elements such as atmospheric circulation, hydrometeorological elements, and physical geography, it is a weakly correlated highly complex nonlinear dynamical system whose interannual variation is not moving in a fixed cycle but contains various time-scale variations and local fluctuations, and this characteristic leads to the difficulty and low accuracy in predicting precipitation in the medium and long term. MEEMD is used to decompose the precipitation data into different decomposition terms and to select PSO-SVM, CNN, and RNN for prediction according to the characteristics of different decomposition terms to reduce the prediction errors caused by the non-smoothness of the data [17].

#### 3.2.1. PSO-SVM

In the PSO algorithm, it is assumed that there are *N* particles in *D*-th dimensional space, $x_{i}=(x_{i1},x_{i2},\cdots x_{id})$ denotes the position of particle i, $v_{i}=(v_{i1},v_{i2},\cdots ,v_{id})$ denotes the velocity of particle *i*, ${pbest}_{id}=(p_{i1},p_{i2}\cdots ,p_{id})$ denotes the best position passed by particle individually, and ${pbest}_{d}=(g_{1},g_{2}\cdots ,g_{d})$ denotes the best position experienced by the population, and in each iteration, the *d*-th dimensional velocity of particle i is updated according to Equation (6) [18]:(6)$$V_{id}^{k+1}=\omega V_{id}^{k}+c_{1}\gamma_{1}({pbest}_{id}-x_{id}^{k})+c2\gamma 2({gbest}_{d}-x_{id}^{k})$$

The *d*-th dimensional position of particle *i* is updated according to Equation (7):(7)$$x_{id}^{k+1}=x_{id}^{k}+V_{id}^{k+1}$$
where $V_{id}^{k}$ denotes the *d*-th dimensional component of the velocity vector of the flight of particle *i* at the d-th iteration, $x_{id}^{k}$ denotes the *d*-th dimensional component of the position vector of particle *i* in the *k*th iteration; $c_{1}$ and $c_{2}$ denote that the learning factors are non-negative constants and take values in the range [1.2, 2], $c_{1}$ is the step used to adjust the particle flying to its optimal position, and $c_{2}$ is the step used to adjust the particle flying to the optimal position of the whole population; $\gamma_{1}$ and $\gamma_{2}$ denote random numbers in the interval [0, 1]; ω is the inertia weight, which describes the effect of the particle’s inertia on the velocity, and its value can regulate the ability of the particle swarm algorithm to find the global and local optimal. $-V^{\max}\leq V_{id}^{k+!}\leq V^{\max}$$-V^{\max}\leq V_{id}^{k+!}\leq V^{\max}$, $V^{\max}$ is a predetermined positive constant that limits the range of velocity variation. The iteration termination condition is generally chosen as the maximum number of iterations or the optimal position searched by the particle swarm so far and satisfying a predetermined minimum adaptation threshold, depending on the specific problem. The values of the parameter penalty coefficient (*c*) and kernel function (*g*) are updated using PSO optimization and are re-substituted into the SVM model for training, its output is saved, and the adaptation values of the particles are calculated using Equation (8).
(8)$$fit(i)={\left(\sum_{i=1}^{l}{\left(y_{i}-\bar{y_{i}}\right)}^{2}\right)}^{\frac{1}{2}},i=1,2,\cdots ,l$$
where fit(i) denotes the adaptation value of the *i*-th particle; l denotes the number of samples; $y_{i}$ denotes the model prediction of the *i*-th particle; $\bar{y_{i}}$ denotes the model expectation of the *i*-th sample [19].

SVM is a supervised learning model commonly used in pattern recognition, classification, and regression analysis. In recent years, SVM has shown many unique advantages in solving problems of small samples, nonlinearity, high-dimensional pattern recognition, etc. It can avoid the problems of overfitting and local minima that often occur in neural networks, has a small generalization error, has a good generalization ability, and has been widely used to solve some regression problems. The particle swarm optimization algorithm is an evolutionary computation technique whose basic idea is to find the optimal solution through information transfer and information sharing among individuals in a population [20,21].

The basic idea of the SVM method is to map the data x in the input space to a high-dimensional feature space through a predefined nonlinear mapping $\Phi :R^{n}\to R^{m}(m\geq n)$, and then do linear regression in this space. Given a data point set of $\left\{(x_{i},y_{i}),i=1,2,\cdots ,N\right\}$, where $x_{i}\in R^{n}$ is the input vector, i.e., the regional rainfall historical time series data, $y_{i}\in R^{n}$ is the output vector corresponding to $x_{i}$, i.e., the rainfall forecast data, and N is the total number of data points, the SVM performs functional regression estimation by Equation (9).
(9)f(x)=ω⋅Φ(x)+b
where ω and Φ(x) are m-dimensional vectors; “⋅” denotes the dot product in the feature space; b∈R is the threshold value [22].

Therefore, the particle swarm algorithm is used to find the optimal solution of the parameter penalty look (*c*) and kernel function (*g*) in the SVM model and arriving at the optimized SVM can effectively solve the shortcomings of small samples, nonlinearity, high dimensionality, and local minimum, while PSO has the advantages of simple operation and fast convergence [23].

#### 3.2.2. CNN

CNN, as one of the artificial neural networks, has become a hot research topic in the field of deep learning and the features of weight sharing and local window sliding make it better to simulate biological neural networks. The model has been improved by continuous machine learning and knowledge accumulation, so that the accuracy of the CNN in predicting sequence specificity has been significantly improved [24].

(1) The input data of the convolutional layer is the output data of the input layer or sampling layer and each convolutional kernel and the feature map of the previous layer perform the convolutional calculation to obtain the feature map of this layer.
(10)$$x_{j}^{l}=f\left[\sum_{i\in Pj}x_{i}^{l-1}\times w_{ij}^{l}+b_{j}^{l}\right]$$
where f(~) is the nonlinear mapping Relu activation function, which is calculated as follows:(11)$$f(u)=\left\{\begin{array}{l} u,\;ifu>0\\ 0,\;otherwise \end{array}\right.$$

Pj represents the selected window on the input feature, i.e., the position on the input feature corresponding to the current convolution kernel at the time of computation during the convolution process; $x_{i}^{l-1}$ and $x_{j}^{l}$ are the corresponding values on the input feature at layer l−1 and the output feature at layer l, respectively; $w_{ij}^{l}$ is the weight value of the convolution kernel; $b_{j}^{l}$ is the bias of the feature, one corresponding to each layer [25].

(2) The fully connected layer can be regarded as a simple single layer perceptron that performs the fully connected computation of all the data in the previous layer. The calculation formula is:(12)$$x^{l}=f(u^{l}),u^{l}=w^{l}*x^{l-1}+b^{l}$$
where f(~) is the nonlinear mapping Relu activation function, $x^{l}$ is the output value of the fully connected layer, and $w^{l}$ and $b^{l}$ are the weight values and biases when computing the l−1-th layer to the *l*-th layer [26].

(3) The output layer (regression layer): assuming that the regression problem corresponds to the *i*-th input feature $x_{i}$ labeled $y^{i}=(y_{1},y_{2},\cdots ,y_{m}{)}^{T}$, *M* is the total dimensionality of the marker vector, then $l_{t}^{i}$ denotes the prediction error (also called residual) between the network regression prediction ${\hat{y}}_{t}^{i}$ on sample i and the true value $y_{t}^{i}$ marker in the *t*-th dimension [27].
(13)$$l_{t}^{i}=y_{t}^{i}-{\hat{y}}_{t}^{i}$$

The l2 loss function for *N* samples is defined as follows:(14)$$Ll2loss=\frac{1}{N}\sum_{i=1}^{N}\sum_{t=1}^{M}(l_{t}^{i}{)}^{2}$$

#### 3.2.3. RNN

The RNN has a good memory capability and can well portray the influence of one moment on subsequent moments, which is especially suitable for dealing with problems that need to consider time-dependent factors. Based on the current state of research in product price analysis and prediction, this paper will use the RNN algorithm to solve the current shortcomings by modeling the highly nonlinear precipitation input and output layers to predict and analyze the future situation. For the time series prediction model, the input and output data are correlated and suitable for the RNN model with memory. Let *x*, *s*, and *o* be 3 vectors that denote the values of input layer, hidden layer, and output layer. *U*, *V*, and *W* denote the weight matrices of input layer and hidden layer, hidden layer and output layer, and hidden layer and hidden layer, respectively:(15)$$st=f(Ux_{t}+Ws_{t}-1+b_{s})$$
(16)$$O_{t}=g(Vs_{t}+bo)$$
where $s_{t}$ is the hidden state at moment *t*; $x_{t}$ is the input at moment *t*; $s_{t-1}$ is the hidden state at moment *t* − 1 and $O_{t}$ is the output at moment *t*; *f* and *g* are the excitation functions of the output and hidden layers, and *b*s and *b**o* are the biases of the output and hidden layers [28,29,30].

### 3.3. Evaluation of Prediction Results

#### 3.3.1. Prediction Accuracy Evaluation

Some of the measured data are used as training samples to rate the parameters of each algorithm and the rest of the measured data are used as test samples and prediction results to compare the prediction effect of each algorithm. In this paper, absolute percentage error (*e*), mean relative error (*MAPE*), root mean square error (*RMSE*), and consistency index (α) are used to evaluate the simulation accuracy. The smaller the *MAPE* and *RMSE*, the higher the simulation accuracy; the larger the α, the closer to 1, the higher the simulation accuracy [31].


(1)Calculate the *e* according to Formula (17).
(17)$$e=\frac{\left|R_{i}-O_{i}\right|}{O_{i}}\times 100\%$$(2)Calculate the *MAPE* according to Formula (18).
(18)$$MAPE=\frac{1}{n}\sum_{i=1}^{n}\frac{\left|R_{i}-O_{i}\right|}{O_{i}}$$(3)Calculate the *RMSE* according to Formula (19).
(19)$$RMSE=\sqrt{\frac{\sum_{i=1}^{n}{\left(R_{i}-O_{i}\right)}^{2}}{n}}$$(4)Calculate the α according to Formula (20).
(20)$$\alpha =1-\frac{\sum_{i=1}^{n}\left|R_{i}-O_{i}\right|}{\sum_{i=1}^{n}\left(\left|R_{i}-O^{\prime }\right|+\left|O_{i}-O^{\prime }\right|\right)}$$
where $O_{i}$ is the actual value, $R_{i}$ is the test value, n is the number of samples, and $O^{\prime }$ is the actual mean value. The smaller the relative error and the root mean square error are, the higher the test accuracy is. The larger the consistency index and effective coefficient are, and the closer they are to 1, the higher the test accuracy is [32].


#### 3.3.2. Z-Index Drought Evaluation

Z-index is one of the most widely used indicators. To calculate the Z index, it is assumed that the precipitation at a certain time period obeys the Person-III type distribution, and by normalizing the precipitation, then its probability density function can be converted by conversion operation. The Z index method, by assuming the rainfall at a certain time period, obeys the Person-III type distribution, and as the time series grows, the precipitation usually obeys normal distribution or close to normal distribution. When calculating the Z value, the skewness coefficient is used. When the standard variable ϕi is determined, the Z value mainly depends on the skewness coefficient Cs, and the Z value is not only related to the precipitation amount but also related to the precipitation distribution characteristics of the area. (1) Z > 1.645, heavy flooding; (2) 1.037 < Z ≤ 1.645, heavy flooding; (3) 0.842 < Z ≤ 1.037, partial flooding; (4) −0.842 ≤ Z ≤ 0.842, normal; (5) −1.037 ≤ Z ≤ −0.842, partial drought; (6) −1.645 ≤ Z ≤ −1.037, heavy drought; (7) Z < −1.645, severe drought [31].

## 4. Validation of the Prediction Model

### 4.1. Overview of the Study Area

The Wu River is the largest tributary on the right bank of the upper reaches of the Yangtze River and the largest river in Guizhou, originating in Yindong Village, Furshan Township, and Weining County, at the eastern foot of the Wumeng Mountains in northwestern Guizhou Province, called the Sanji River. The Wujiang River basin spans 41 counties (cities and districts) in eight cities (states) in western, central, and northeastern Guizhou. In the west, the watershed with Hengjiang River and Niujiang River is the Wumeng Mountain branch; in the south, the watershed with Hongshui River and its tributary Beipanjiang River is the Wumeng Mountain and Miaoling Mountain Range; in the northwest, the watershed with Chishui River and Qijiang River is the Dalou Mountain Range; in the east, the watershed with Yuanjiang River system is the Wuling Mountain Range; in the northeast, the watershed is adjacent to Yangtze River and Hubei Qing Shuijiang River. The watershed area of the Wujiang River basin in Guizhou is 66,807 km^2^, accounting for 76% of the total watershed area. Therefore, the analysis and prediction of annual precipitation in the Wujiang River basin is highly representative of the region and Figure 2 shows the water system map of the Wujiang River basin.

### 4.2. Data and Methods

#### 4.2.1. Data Sources and Processing

The precipitation data in this paper were obtained from 58 years (1961–2018) of precipitation monitoring data from 17 ground stations in the Wujiang River basin compiled by the Guizhou Provincial Meteorological Bureau, which were accurate and credible. In terms of data processing, the day-by-day data were firstly counted to obtain the multi-year average and chronological average for each month and season, respectively.

#### 4.2.2. Data Overview

As shown in Figure 3, the multi-year average precipitation in the Wujiang River basin in the summer from 1961 to 2018 was 518.66 mm. The minimum value occurred in 1972 with 256.92 mm and the maximum value occurred in 1998 with 676.36 mm. According to the trend value curve, it can be seen that the average annual precipitation in the Wujiang River basin showed an increasing trend, with an average annual increase of 0.5691 mm. According to the 5-year sliding average curve, it can be seen that the average annual summer precipitation in the Wujiang River basin has roughly undergone a cycle of abundance → depletion → abundance → depletion over the 51 years from 1961 to 2018.

### 4.3. Implementation of the Prediction Model

#### 4.3.1. MEEMD Decomposition of Precipitation Time Series

The quarterly precipitation series of the Wujiang River basin has obvious nonlinearity and non-smoothness. In order to better analyze the precipitation time series locally to obtain higher prediction accuracy, we used MEEMD to decompose the precipitation sample data of the Wujiang River basin for a total of 58 summers from 1961–2018. In the decomposition, the white noise intensity is set to 0.2, the maximum number of intrinsic modes is 6, the number of joining noise is 30, the embedding dimension is 6, and the time delay is 1. When the alignment entropy of the signal θ_0_ = 0.55, the original signal x of precipitation is decomposed into IMF1~IMF3 and the residual term (RS4). When the alignment entropy of the signal is taken as θ_0_ = 0.6, the decomposition is IMF1~IMF4 and the residual term (RS5). Figure 4 shows the results of MEEMD decomposition of precipitation.

It was calculated that the *MAPE* of the reconstructed signal and the original signal for the MEEMD four-layer decomposition was 0.17, while the *MAPE* of the reconstructed signal and the original signal for the MEEMD five-layer decomposition was 1.95 × 10^−17^, indicating that the actual components of the MEEMD five-layer decomposition and the original signal were in good agreement; thus, the five-layer decomposition was used in this paper. Taking the summer precipitation of the Wujiang River basin in Guizhou Province from 1961–2018 as an example, the first 38 years of data to train the network and the last 20 years of data to test the network, the decomposition terms and based on the root mean square error index to filter, IMF1, IMF2 decomposition terms change faster with time, for the high frequency part of the summer precipitation signal, will be predicted using the PSO-SVM. The IMF3 and IMF4 decomposition terms change slowly with time and are the low-frequency part of the summer precipitation signal with strong periodicity, which will be predicted by CNN, while the residual term RS5 will be predicted by the RNN model.

#### 4.3.2. PSO-SVM Prediction

Parameter initialization: learning factor (*c*1 = 1.5; *c*2 = 1.7); maximum number of evolutions (maxgen) = 200; maximum number of populations (sizepop = 20); initial value of k (*k* = 0.6), relationship between rate and *x* (*v* = *kX*); elasticity coefficient in front of velocity in the rate update formula (*wV* = 1); velocity in front of the population update formula elasticity coefficient (*wP* = 1); cross validation parameter (*v* = 5); maximum value of the variation of parameter c (popcmax = 100); minimum value of the variation of parameter *c* (popcmin = 1); maximum value of the variation of parameter *g* (popgmax = 0.5); minimum value of the variation of parameter *g* (popgmin = 0.3). The population size was (pop = 500). Figure 5 shows the comparison between the predicted and actual values of IMF1~IMF2, where Figure 5a,b shows that the predicted values of the IMF1 and IMF2 subsets are very close to the actual values, which verifies the validity of the PSO-SVM model. Figure 5c,d shows the well-trained adaptation functions of IMF1 and IMF2, respectively. Among them, the IMF1 trained optimal penalty factor *c* = 1, kernel function *g* = 100, and fitness CVmse = 0.235533; the IMF2 trained optimal penalty factor *c* = 1, kernel function *g* = 8, and fitness CVmse = 0.1275.

#### 4.3.3. CNN Prediction

The CNN is constructed as three layers: one input layer, one convolutional layer, and one fully connected layer. The input samples are 55 sets of 2 × 2 × 1 3D data, the convolutional layer is set up as four convolutional kernels for convolutional operation, respectively, and the size of the convolutional kernels is 1 × 1. The output of the convolutional layer is used as the input of the fully connected layer, which constitutes a single layer perceptron and outputs the prediction values.
(1)Input layer: the size of the input layer is 2 × 2 × 1;(2)convolution layer: since the input data matrix is small, a smaller 1 × 1 convolution kernel is used. The number of convolution kernels is four and the step size is one. In order to prevent the matrix from losing too much information in the convolution process, the Padding=‘same’ method is used to make up the zeroes. At the same time, the step length of convolution is set to one;(3)batch normalization layer: the mean−variance method is used to batch normalize;(4)activation function layer: the reluLayer activation function is used. The constructed CNN model is shown in Figure 6.

The training CNN neural network parameters are set as follows: trainingOptions: solverName is the stochastic gradient descent (sgdm) of the driving volume, initial learning rate = 0.01, LearnRateDropFactor = 10^−3^; LearnRateDropPeriod = 20; MiniBatchSize = 64; MaxEpochs = 4000; the first 35 sets of data were used for training the network, and the last 20 sets of data were used for testing. Figure 7 shows the CNN network model and the comparison of IMF3−IMF4. It can be seen that the predicted value of the trained model using CNN is very close to the actual value, which verifies the effectiveness of the CNN model.

#### 4.3.4. RNN Prediction of the Residual Term RS5

The RNN model is used to predict the decomposition term RS5. The original sample data is normalized using the maximum–minimum method, the sample data structure is 58 × 1 times the series data, and the data series is processed into 55 × 10 by the [in,out] = create_serino (lie_samp,10) statement, and the first 10 columns of sample data are selected for correlation analysis; it is found that the correlation coefficient of the first 9 columns of sample data is greater than 0.5, which is a strong correlation, that is, when 50 years can represent the general distribution of the whole series. Given a precipitation time series with input data $x_{n}=(x_{1},x_{2},\cdots x_{t},\cdots x_{49})$ and output data $y=x_{50}$, the sequence goes through a cycle of roughly 5 years, and the input data is further processed into 9 × 5 × 49 by data preprocessing, normalization, and structuring, and the output layer is 9 × 1. The network structure is established as dimensions, with the number of neurons in the input layer being 9, the number of neurons in the hidden layer being 10, and the number of neurons in the output layer being 1. The activation function is selected as sigmoid, learning rate = 0.001, training step = 5000, batch = 20, and the learning rate algorithm is RAdam. Figure 8 shows the structure of the RNN and Figure 9 shows the comparison between the predicted and actual values of RS5.

### 4.4. Evaluation and Analysis of Simulation Results

Algorithms such as BP, SVM, PSO−SVM, CNN, RNN, and MDPRM were selected to train the summer precipitation data of the first 38 years from 1961–2018 in the Wujiang River basin with the same initial value settings; the latter 20 sets of data were tested, and the simulation results and the measured data were compared and analyzed. Figure 10 shows the comparison of the results of the simulation of summer precipitation changes in the Wujiang River basin.

Overall, the passing rate of the six forecast models is 100% (the absolute percentage error is less than 20% as the passing standard). From Figure 10, it can be intuitively seen that the prediction results of the combined MEEMD model in this paper are closer to the true values and have better prediction accuracy. In order to evaluate the model performance more comprehensively, the statistical values of *MAPE*, *RMSE*, and α are given in Table 2. As can be seen in Table 2, the *MAPE* of MEEMD decreases from 0.31 to 0.09, *RMSE* decreases from 0.56 to 0.30, and α increases significantly from 0.33 to 0.86. It can be seen that the MDPRM model is significantly better than the other five models, with a more accurate time−series data prediction capability.

### 4.5. Application of Precipitation Prediction Results

The MEEMD decomposition of 58 years of summer precipitation in the Wujiang River basin from 1961–2018 was used to predict the precipitation $\mathrm{IMF}1^{\prime }$, $\mathrm{IMF}2^{\prime }$, $\mathrm{IMF}3^{\prime }$, ${\mathrm{IMF}4}^{\prime }$, and $\mathrm{RS}5^{\prime }$ for the next 10 years using a combined prediction model, and the precipitation prediction value S was obtained by accumulating $S=\mathrm{IMF}1^{\prime }+\mathrm{IMF}2^{\prime }+\mathrm{IMF}3^{\prime }+{\mathrm{IMF}4}^{\prime }+\mathrm{RS}5^{\prime }$. The prediction results are shown in Table 3. According to the prediction results, Z indicators are used to analyze the drought level characteristics of the Wujiang River basin in the next 10 years. According to the MDPRM prediction results, the average precipitation of summer precipitation in the Wujiang River basin is 466.42 mm in 2019–2028, the minimum precipitation is 440.94 mm in 2020, and the maximum precipitation is 497.94 mm in 2024. Further using Z indicators, the assessment of the agricultural drought level shows that the summer season is normal during this 10-year period.

## 5. Conclusions


(1)By setting the MEEMD method with the white noise intensity of 0.2, the maximum number of intrinsic modes of 6, the number of times to join the noise of 30, the embedding dimension of 6, the time delay of 1, and the PE of the signal taken as 0.6, the original summer precipitation signal x of the Wujiang River basin for 58 years (1961–2018) is decomposed into IMF1-IMF4 and the residual term RS5, which can accurately identify hierarchical features of precipitation data, effectively suppress the modal mixing phenomenon in the EMD decomposition process, and reduce the reconstruction error.(2)In this paper, a new precipitation MDPRM is proposed, i.e., according to the prediction accuracy and prediction efficiency, the PSO−SVM is used to predict IMF1 and IMF2 decomposition terms, CNN is used to predict IMF3 and IMF4 decomposition terms, and RNN is used to predict the residual term RS5, and the precipitation prediction values are obtained by accumulating the prediction values of each sub−term. Compared with BP, SVM, PSO-SVM, CNN, and RNN, *MAPE* is reduced from 0.31 to 0.09, *RMSE* is reduced from 0.56 to 0.30, and α is significantly increased from 0.33 to 0.86, indicating that the MDPRM constructed in this paper improves the prediction accuracy.(3)The MDPRM constructed in this paper was used to predict the summer precipitation in the Wujiang River basin for the next 10 years from 2019 to 2028 and the predicted results showed that the average precipitation for the next 10 years was 466.42 mm, with a minimum of 440.94 mm in 2020 and a maximum of 497.94 mm in 2024. Further assessment of the agricultural drought rating using the Z indicator showed that the summers were normal during the 10 years. The results of the precipitation prediction and drought characteristics analysis provide technical support for the scheduling and efficient use of water resources in the summer to maximize economic, social, and ecological benefits.


## Figures and Tables

**Figure 1 sensors-22-06415-f001:**
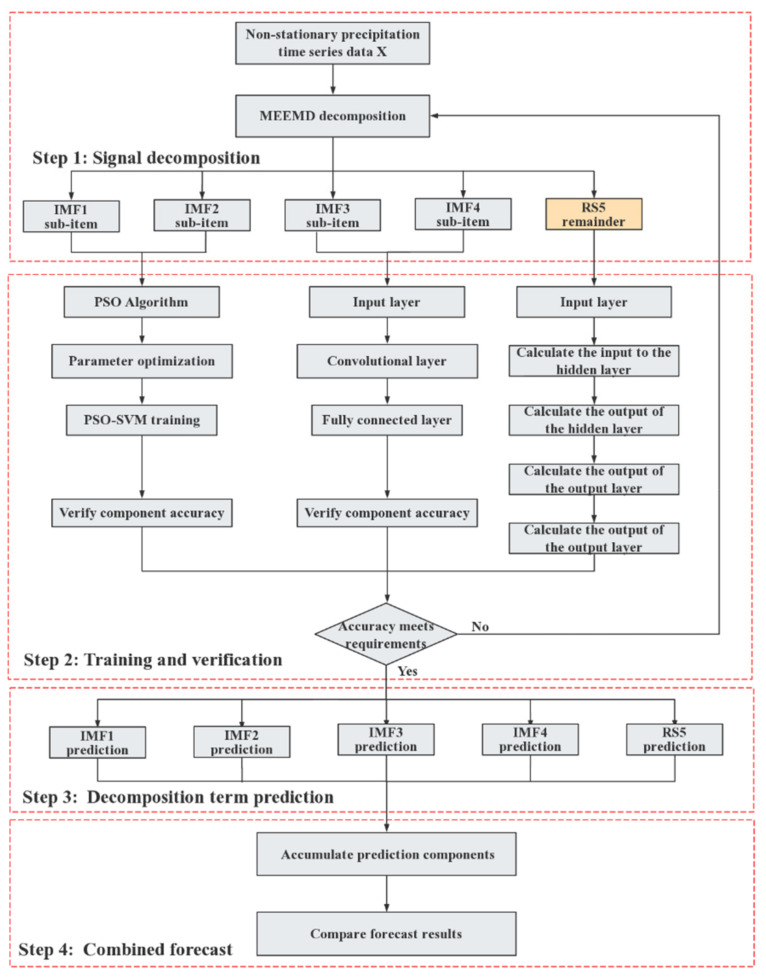
The MDPRM model structure.

**Figure 2 sensors-22-06415-f002:**
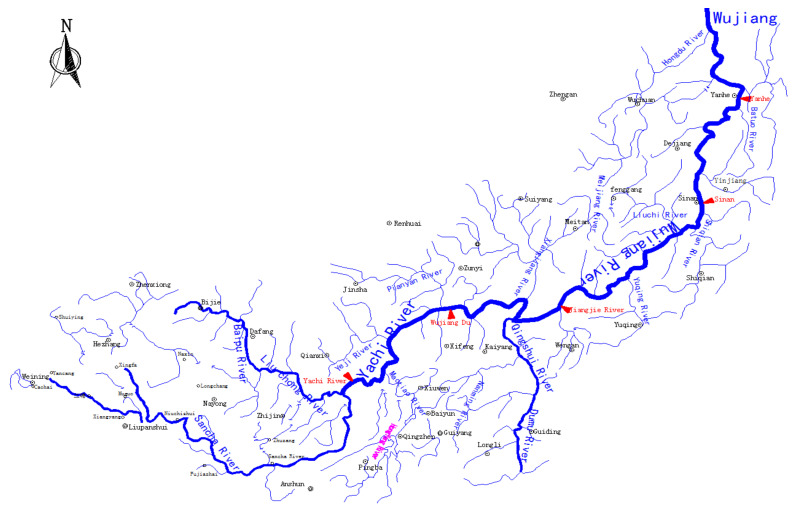
River system map of the Wujiang River Basin.

**Figure 3 sensors-22-06415-f003:**
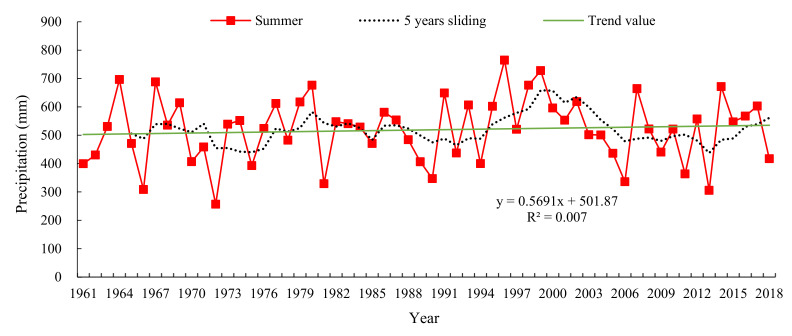
Seasonal precipitation in the Wujiang River Basin (1986–2018).

**Figure 4 sensors-22-06415-f004:**
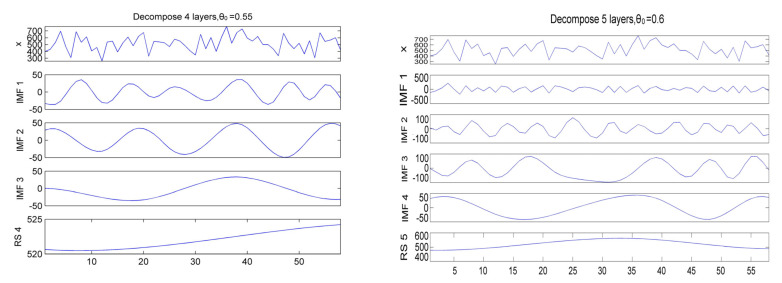
Results of MEEMD decomposition of summer precipitation in the Wujiang River basin.

**Figure 5 sensors-22-06415-f005:**
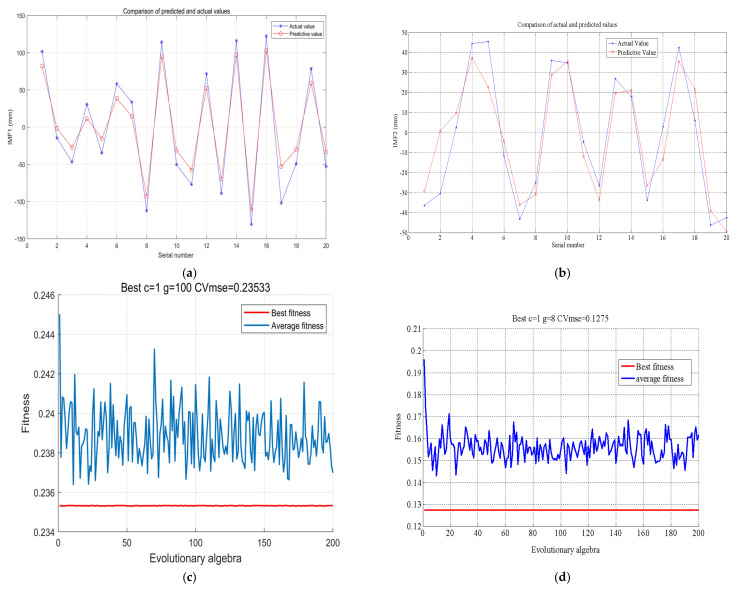
Comparison of predicted and actual values of IMF1−IMF2. (**a**) IMF1 predicted versus true values; (**b**) IMF2 predicted versus true values; (**c**) IMF1 fitness value; (**d**) IMF2 adaptation value.

**Figure 6 sensors-22-06415-f006:**
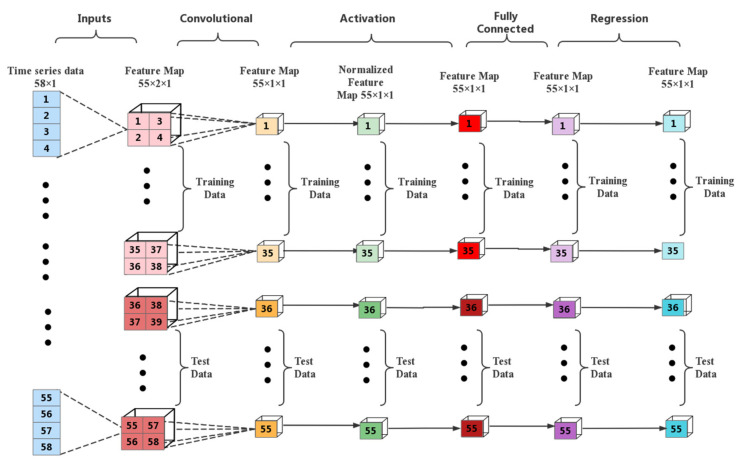
CNN model.

**Figure 7 sensors-22-06415-f007:**
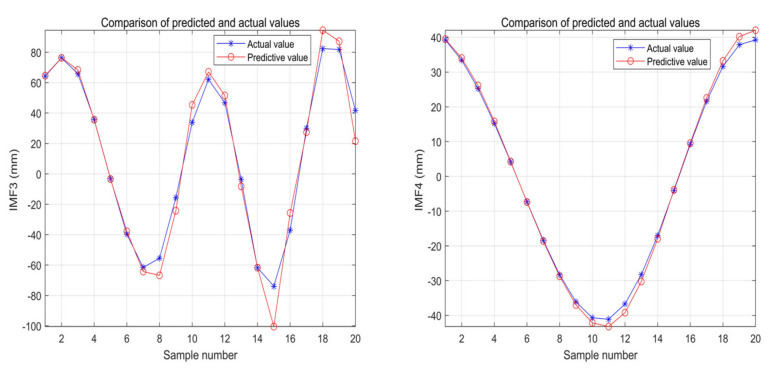
Comparison of IMF3−IMF4 predicted true values.

**Figure 8 sensors-22-06415-f008:**
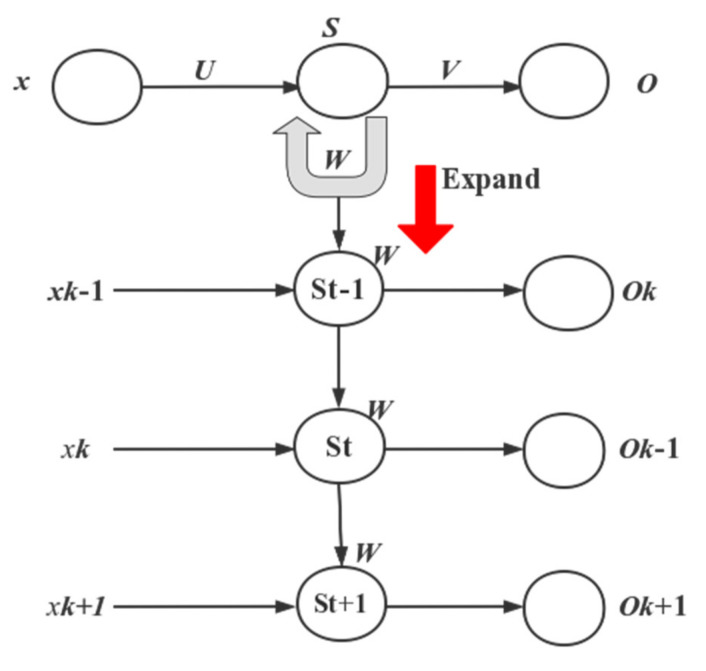
The structure of RNN.

**Figure 9 sensors-22-06415-f009:**
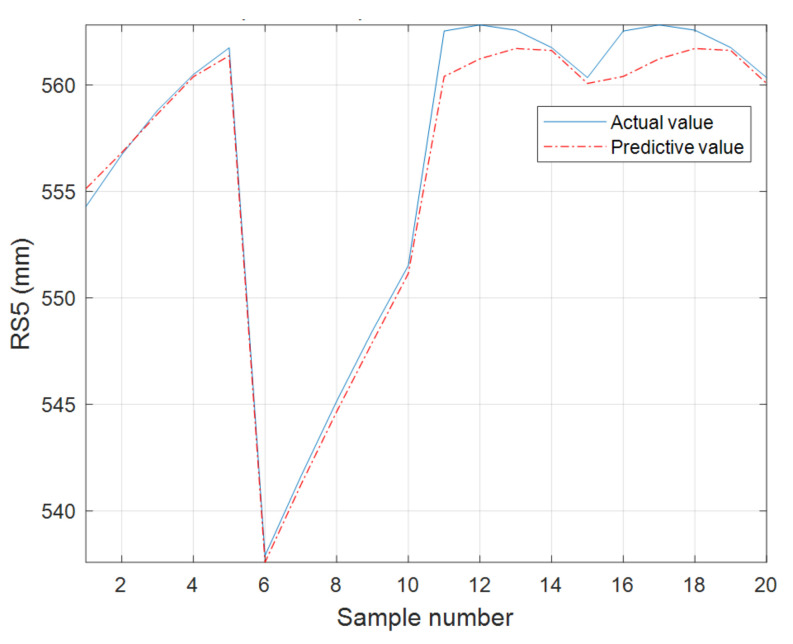
Comparison of the predicted and actual values of RNN.

**Figure 10 sensors-22-06415-f010:**
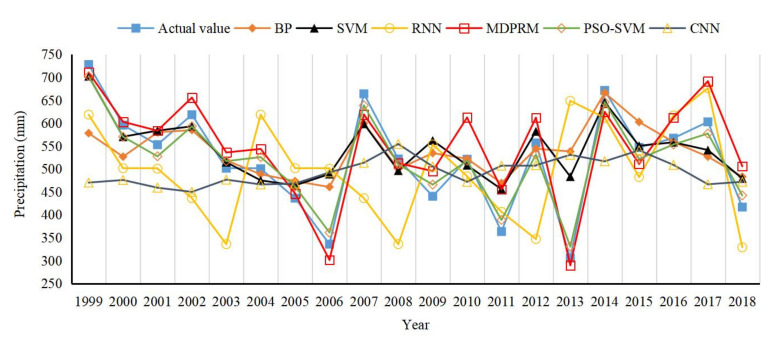
Comparison of summer precipitation test results in the Wujiang River basin from 1999–2018.

**Table 1 sensors-22-06415-t001:** Study of correlation portfolio prediction methods.

Serial Number	Researchers	Model Algorithm	Effect	Prediction Scenarios	Research Time
[1]	Zhang et al.	MEEMD-ARIMA	Better than CEEMD-ARIMA model or EEMD-ARIMA model	Annual runoff in the lower reaches of the Yellow River	2020
[2]	Wu et al.	Improved Grey BPNN-MEEMD-ARIMA	The effectiveness of the proposed wave energy forecast model is validated.	Prediction of wave energy	2021
[3]	Wang et al.	MEEMD-CS-Elman	Better than EMD-Elman or MEEMD-Elman model	Gold futures AU0 price data	2021
[4]	Song et al.	MEEMD-SAE-Elman	Better than Elman, EMD-Elman, EMD-SAE-Elman	Wind power ultra-short-term	2020
[5]	Xie et al.	MEEMD+WLS-SVM	Better than Wavelet Neural Network, Support Vector Machine, Least Squares Support Vector Machine	Landslide Deformation Prediction in Danba	2017
[6]	Lin et al.	LSTM-MEEMD	Better than MLP, SVR and SL	Precious Metals Price	2022
[7]	Yang et al.	MEEMD-LSTM. and IWOA	Better than 11 other benchmark models	Carbon price	2020
[8]	Yang et al.	MEEMD-LSTM	Better than LSTM models and one-factor MEEMD-LSTM	Carbon price	2022

**Table 2 sensors-22-06415-t002:** Evaluation of simulation accuracy of different model algorithms.

Serial Number	Model Algorithm	*MAPE*	*RMSE*	α
1	BP	0.14	0.37	0.64
2	SVM	0.12	0.35	0.75
3	PSO−SVM	0.29	0.54	0.45
4	CNN	0.32	0.57	0.33
5	RNN	0.26	0.51	0.53
6	MDPRM	0.09	0.30	0.86

**Table 3 sensors-22-06415-t003:** Precipitation forecast and drought assessment for the summer of 2019–2028 in the Wujiang River basin.

Decomposition Items	Prediction Method	Years									
		2019	2020	2021	2022	2023	2024	2025	2026	2027	2028
$\mathrm{IMF}1^{\prime }$	PSO−SVM	−0.09	−0.09	−0.09	−0.09	−0.09	−0.09	−0.09	−0.09	−0.09	−0.09
$\mathrm{IMF}2^{\prime }$	PSO−SVM	5.09	−7.04	−6.85	0.11	−6.15	2.03	2.44	1.85	−5.38	−2.11
$\mathrm{IMF}3^{\prime }$	CNN	−53.30	−64.53	−50.68	−24.92	−1.26	10.70	9.25	−0.54	−9.73	−10.83
${\mathrm{IMF}4}^{\prime }$	CNN	33.07	27.04	19.43	9.54	0.47	−8.58	−17.32	−24.53	−30.55	−35.04
$\mathrm{RS}5^{\prime }$	RNN	488.51	485.56	485.77	489.00	493.07	493.87	493.01	493.18	489.20	492.24
Predicted Results (mm)		473.29	440.94	447.58	473.64	486.05	497.94	487.30	469.87	443.45	444.18
Z-index		−0.35	−0.65	−0.59	−0.34	−0.23	−0.12	−0.22	−0.38	−0.63	−0.62
Drought Level		Normal	Normal	Normal	Normal	Normal	Normal	Normal	Normal	Normal	Normal

## Data Availability

Requests for materials should be addressed to wyt409011805@hnu.edu.cn (Y.W.) or liujian@hnu.edu.cn (J.L.).

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
