# Peer review of "MEEMD Decomposition–Prediction–Reconstruction Model of Precipitation Time Series"

_sensors, 2022, doi:10.3390/s22176415_

Round 1

Reviewer 1 Report

A brief summary 

We see the paper's primary goal as the authors' effort to expand knowledge and support solutions to strengthen theoretical knowledge, but especially to obtain and prove practical prediction tools to solve the problem of low accuracy of precipitation time series data prediction.

The main contribution and strength of the article are that the article moves the discussion to the issue by using a combination of prediction methods, which have their own characteristics and conditions of use, in favor of simplifying prediction processes, accuracy, and prediction efficiency. The developed proposal takes advantage of the prediction models to perform the prediction of different components separately and finally combines each prediction component to produce the final prediction result, which provides a new method to improve the accuracy of regional precipitation forecast.

General concept comments

After reading the abstract of the paper, it may not be clear to the reader what research question the authors were solving.

In the article abstract: Clearly identify the research question or hypothesis addressed within the broader context and purpose of the article.

Following the abstract of the paper, a specific research question/hypothesis should be defined in the Introduction section, which will be verified on the basis of experimental results and validation of the prediction model within the research of the issue. Finally, the main objective of the work is briefly stated and the main conclusions are highlighted.

In the Introduction section: Clearly define the research question/hypothesis of your work.

The literature search on the issue is adequate.

The authors used the relevant methodology for solving the problem and the extensive database of the Wujiang 584 River basin for 58 years (1961-2018).

The tables and figures correlate with the content of the paper, they are easy to interpret and understand.

The cited references (37 sources) are correct in my opinion.

The article has its limits, but it contributes to further creative discussion of the issue with practical use.

The conclusions are clearly formulated, in accordance with the presented evidence and arguments of the authors.

We can consider that the focus of the paper and the selected knowledge can be adequately extracted for other application areas with the required prediction.

The paper has the potential to generate additional research questions for further scientific work.

The manuscript is clear, relevant for the field, and presented in a required structured manner.

Specific comments 

In line 57:

Is the beginning of a sentence missing or did the authors want to cite a source?

„ ) can identify the hierarchical characteristics of precipitation data more accurately, which can effectively reduce the interference between the decomposition terms, reduce the complexity of prediction, select a suitable prediction method for each decomposition term, and improve the overall prediction accuracy and prediction efficiency of the combined prediction method with more obvious advantages [2].“.

Overall Recommendation

After minor corrections, I recommend the paper for publication as part of the academic and research discussion on the given topic.

Author Response

Dear editors and reviewers:

Thank you very much for your comments concerning our manuscript entitled “MEEMD Decomposition-Prediction-Reconstruction Model of Precipitation Time Series” (Manuscript ID: sensors-1847938). Your comments are so valuable and helpful for us to correct and improve our manuscript. We have tried our best to revise and improve the manuscript, and given an explanation according to each issue of your report. The revised portion is marked in red in the revised manuscript. The main corrections and responses to your comments are listed in the appendix.

Yours sincerely,

Yongtao Wang, Jian Liu, Rong Li, Xinyu Suo and EnHui Lu

Appendix 1: Our responses to the reviews’ comments

  • In line 57:

Is the beginning of a sentence missing or did the authors want to cite a source?

„ ) can identify the hierarchical characteristics of precipitation data more accurately, which can effectively reduce the interference between the decomposition terms, reduce the complexity of prediction, select a suitable prediction method for each decomposition term, and improve the overall prediction accuracy and prediction efficiency of the combined prediction method with more obvious advantages [2].“.

Response:Thank you very much for your careful and valuable comments. I apologize for an error here. I have double-checked the entire sentence. I have reorganized the abstract to more clearly describe the main points of the study, and the optimized abstract is as follows:

Abstract: To address the problem of low prediction accuracy of precipitation time series data, an improved overall mean empirical modal decomposition-prediction-reconstruction model (MDPRM) is constructed in this paper. First, the non-stationary precipitation time series are decomposed into multiple decomposition terms by the improved overall mean empirical modal decomposition (MEEMD). Then, particle swarm optimization support vector machine (PSO-SVM), convolutional neural network (CNN) and recurrent neural network (RNN) models are used to make predictions according to the characteristics of different decomposition terms. Finally, the prediction results of each decomposition term are superimposed and reconstructed to form the final prediction results. And the application is carried out with the summer precipitation in the Wujiang River basin of Guizhou Province from 1961 to 2018, using the first 38 years of data to train MDPRM and the last 20 years of data to test MDPRM, and comparing with feedback neural network (BP), support vector machine (SVM), particle swarm optimization support vector machine (PSO-SVM), convolutional neural network (CNN) and recurrent neural network (RNN), etc. The results show that the mean relative error (MAPE) of the proposed MDPRM is reduced from 0.31 to 0.09, the root mean square error (RMSE) is reduced from 0.56 to 0.30, and the consistency index (α) is significantly improved from 0.33 to 0.86, which has higher prediction accuracy. Finally, the trained MDPRM predicts the average summer precipitation in the Wujiang River basin from 2019 to 2028 to be 466.42 mm, the minimum precipitation in 2020 to be 440.94 mm, and the maximum precipitation in 2024 to be 497.94 mm. Based on the prediction results, the agricultural drought level is evaluated using the Z index, which indicates that the summer is normal in the 10-year period. The study provides technical support for the effective guidance of regional water resources allocation and scheduling and drought mitigation.

Reviewer 2 Report

There are some suggestions to be revised to improve the quality of this paper as follows:

1.              Abstract should be clear and check whether all the points mentioned in abstract are addressed in this manuscript. The abstract must summarize the performance evaluation results.

2.              The authors should provide solid motivation for their work based on the existing literature. In addition, the main contributions should be defined as set of bullets at the end of introduction section.

3.              A table should be added to summarized related work and state the approach and result achieved by other research in the introduction.

4.              The paper contains only references before 2022; other years references also need to be added especially of 2021 and 2022.

5.              English usage needs to be improved. Please carefully proofread the paper to get rid of typos and grammar mistakes.

6.              References should follow the template. Please correct them all.

Author Response

Dear editors and reviewers:

Thank you very much for your comments concerning our manuscript entitled “MEEMD Decomposition-Prediction-Reconstruction Model of Precipitation Time Series” (Manuscript ID: sensors-1847938. Your comments are so valuable and helpful for us to correct and improve our manuscript. We have tried our best to revise and improve the manuscript, and given an explanation according to each issue of your report. The revised portion is marked in red in the revised manuscript. The main corrections and responses to your comments are listed in the appendix.

Yours sincerely,

Yongtao Wang, Jian Liu, Rong Li, Xinyu Suo and EnHui Lu

Appendix 2: Our responses to the reviews’ comments

  • Abstract should be clear and check whether all the points mentioned in abstract are addressed in this manuscript. The abstract must summarize the performance evaluation results.

Response:Thank you very much for your careful and valuable comments. As you commented, abstract should be clear and check whether all the points mentioned in abstract are addressed in this manuscript. The abstract must summarize the performance evaluation results, which We have modified as follows:

Abstract: To address the problem of low prediction accuracy of precipitation time series data, an improved overall mean empirical modal decomposition-prediction-reconstruction model (MDPRM) is constructed in this paper. First, the non-stationary precipitation time series are decomposed into multiple decomposition terms by the improved overall mean empirical modal decomposition (MEEMD). Then, particle swarm optimization support vector machine (PSO-SVM), convolutional neural network (CNN) and recurrent neural network (RNN) models are used to make predictions according to the characteristics of different decomposition terms. Finally, the prediction results of each decomposition term are superimposed and reconstructed to form the final prediction results. And the application is carried out with the summer precipitation in the Wujiang River basin of Guizhou Province from 1961 to 2018, using the first 38 years of data to train MDPRM and the last 20 years of data to test MDPRM, and comparing with feedback neural network (BP), support vector machine (SVM), particle swarm optimization support vector machine (PSO-SVM), convolutional neural network (CNN) and recurrent neural network (RNN), etc. The results show that the mean relative error (MAPE) of the proposed MDPRM is reduced from 0.31 to 0.09, the root mean square error (RMSE) is reduced from 0.56 to 0.30, and the consistency index (α) is significantly improved from 0.33 to 0.86, which has higher prediction accuracy. Finally, the trained MDPRM predicts the average summer precipitation in the Wujiang River basin from 2019 to 2028 to be 466.42 mm, the minimum precipitation in 2020 to be 440.94 mm, and the maximum precipitation in 2024 to be 497.94 mm. Based on the prediction results, the agricultural drought level is evaluated using the Z index, which indicates that the summer is normal in the 10-year period. The study provides technical support for the effective guidance of regional water resources allocation and scheduling and drought mitigation.

  • The authors should provide solid motivation for their work based on the existing literature. In addition, the main contributions should be defined as set of bullets at the end of introduction section.

Response:Thank you very much for your valuable comments. We have described the main contributions of this study at the end of the introductory section, as follows:

To this end, MDPRM is proposed in this paper and applied to the summer precipitation prediction in the Wujiang River basin. The improved integrated empirical modal decomposition (EEMD) algorithm is constructed by improving the integrated empirical modal decomposition (MEEMD) through the randomness detection based on the permutation entropy (PE). First, the original summer precipitation signal x in the Wujiang River basin for 58 years (1961-2018) is decomposed into 5 layers of IMF1-IMF4 with different frequencies and the residual term RS5, and the errors of each decomposition term are checked; second, the prediction methods such as PSO-SVM, CNN, and RNN are preferred to carry out simulations for different decomposition terms that meet the requirements; third, the predicted values of each decomposition term are combined to form Finally, the constructed MDPRM is used to predict the summer precipitation in the Wujiang River basin for the next 10 years (2019-2028) and to evaluate the drought situation. The results show that the MDPRM constructed in this paper significantly improves the prediction accuracy and outperforms BP, SVM, PSO-SVM, CNN and RNN, etc. The MDPRM fully combines the ability of MEEMD to accurately identify multi-level features of nonlinear and nonsmooth signals, and preferably selects PSO-SVM, CNN, RNN and other prediction models for the combination of prediction, which provides a new method to improve the regional precipitation It provides a new way of thinking and method to improve the accuracy of regional precipitation prediction.

  • A table should be added to summarized related work and state the approach and result achieved by other research in the introduction.

Response:Thank you for providing very good revisions, which are very helpful to improve the quality of the paper and to clearly describe the problem. We have added a table at the end of the summary of the relevant work with the methodology and results obtained from the relevant studies, as follows:

 Table 1. Study of correlation portfolio prediction methods

Serial number

Researchers

Model Algorithm

Effect

Prediction Scenarios

Research Time

1

Zhang, X. et al.

MEEMD-ARIMA

Better than CEEMD-ARIMA model or EEMD-ARIMA model

Annual runoff in the lower reaches of the Yellow River

2020

2

Wu, F et al.

Improved Grey BPNN-MEEMD-ARIMA

The effectiveness of the proposed wave energy forecast model is validated.

Prediction of wave energy

2021

3

Wang, X. et al.

MEEMD-CS-Elman

Better than EMD-Elman or MEEMD-Elman model

Gold futures AU0 price data

2021

4

Song et al.

MEEMD-SAE-Elman

Better than Elman, EMD-Elman, EMD-SAE-Elman

Wind power ultra-short-term

2020

5

Xie et al.

MEEMD+WLS-SVM

Better than Wavelet Neural Network, Support Vector Machine, Least Squares Support Vector Machine

Landslide Deformation Prediction in Danba

2017

6

Lin et al.

LSTM-MEEMD

Better than MLP, SVR and SL

Precious Metals Price

2022

6

Yang et al.

MEEMD-LSTM.

and IWOA

Better than 11 other benchmark models

Carbon price

2020

8

Yang et al.

MEEMD-LSTM

Better than LSTM models and one-factor MEEMD-LSTM

Carbon price

2022

  • The paper contains only references before 2022; other years references also need to be added especially of 2021 and 2022.

Response:Thank you for your valuable suggestions, we have updated the references and added mainly relevant research literature for the years 2021 and 2022.

  • English usage needs to be improved. Please carefully proofread the paper to get rid of typos and grammar mistakes.

Response:Thank you for your excellent suggestions, and as you said, we have carefully proofread the paper and done our best to eliminate spelling errors and grammatical mistakes and improve the language and editorial quality of the paper.

  • References should follow the template. Please correct them all.

Response:Thank you for the excellent suggestion, as you said, we have followed the template for citation and have double checked the full text for correction.

Reviewer 3 Report

It is an interesting proposal, an appropriate one, including a correct research methodology and conclusive results. The present year must be taken into account in the interpretation. The forecast, based on the data string, is until the year 2028. In this context, I consider that a re-evaluation of the text is necessary. Also, the introduction should be summarized.

Author Response

Dear editors and reviewers:

Thank you very much for your comments concerning our manuscript entitled “MEEMD Decomposition-Prediction-Reconstruction Model of Precipitation Time Series” (Manuscript ID: sensors-1847938). Your comments are so valuable and helpful for us to correct and improve our manuscript. We have tried our best to revise and improve the manuscript, and given an explanation according to each issue of your report. The revised portion is marked in red in the revised manuscript. The main corrections and responses to your comments are listed in the appendix.

Yours sincerely,

Yongtao Wang, Jian Liu, Rong Li, Xinyu Suo and EnHui Lu

Appendix 3: Our responses to the reviews’ comments

  • It is an interesting proposal, an appropriate one, including a correct research methodology and conclusive results. The present year must be taken into account in the interpretation. The forecast, based on the data string, is until the year 2028. In this context, I consider that a re-evaluation of the text is necessary. Also, the introduction should be summarized.

Response:Thank you for your acknowledgement of the work of this study, which is very important to us. The accuracy of the constructed MDPRM is further verified by our comparison of the predicted summer precipitation in the Wujiang River basin for the last three years 2019, 2020 and 2021 with the actual data, which shows that the prediction pass rate is 100%.

Precipitation prediction can scientifically and rationally guide the planning, development and distribution of regional water resources, reduce flood and drought disaster losses and alleviate water shortages, etc. The prediction results can provide reference for water supply and demand analysis, water supply planning and water resources management.

The MDPRM constructed in this paper was used to predict the summer precipitation in the Wujiang River basin for the next 10 years from 2019 to 2028, and the predicted results showed that the average precipitation for the next 10 years was 466.42 mm, with a minimum of 440.94 mm in 2020 and a maximum of 497.94 mm in 2024. Further assessment of the agricultural drought rating using the Z indicator showed that the summers were normal during the 10 years.. The results of precipitation prediction and drought characteristics analysis provide technical support for the scheduling and efficient use of water resources in summer to maximize economic, social and ecological benefits.

Round 2

Reviewer 2 Report

The authors addressed all comments